# Definition and terminology of developmental language disorders—Interdisciplinary consensus across German-speaking countries

Carina Lüke[1]☯*, Christina Kauschke[2]☯, Andrea Dohmen[3], Andrea Haid[4], Christina Leitinger[5], Claudia Männel[6,7], Tanja Penz[8], Steffi Sachse[9], Wiebke Scharff Rethfeldt[10], Julia Spranger[11], Susanne Vogt[12], Marlen Niederberger[11‡], Katrin Neumann[13‡]

1 Faculty of Human Sciences, Department of Special Education and Therapy in Language and Communication Disorders, University of Würzburg, Würzburg, Germany, 2 Department of German Linguistics, University of Marburg, Marburg, Germany, 3 Department of Applied Health Sciences, Speech and Language Therapy, University of Applied Health Sciences (HS Gesundheit), Bochum, Germany, 4 Swiss University of Speech and Language Sciences SHLR, Rorschach, Switzerland, 5 Logopädieaustria, Professional Association of Austrian Speech-Language Therapists, Vienna, Austria, 6 Department of Audiology and Phoniatrics, Charité –Universitätsmedizin Berlin, Berlin, Germany, 7 Department of Neuropsychology, Max Planck Institute for Human Cognitive and Brain Sciences, Leipzig, Germany, 8 University Hospital for Hearing, Voice and Language Disorders, Innsbruck, Austria, 9 Institute of Psychology, University of Education Heidelberg, Heidelberg, Germany, 10 Faculty of Social Sciences, Logopedics, Hochschule Bremen–City University, Bremen, Germany, 11 Department of Research Methods in Health Promotion and Prevention, Institute for Health Sciences, University of Education Schwäbisch Gmünd, Schwäbisch Gmünd, Germany, 12 Department of Health & Social Affairs, University of Applied Sciences Fresenius Frankfurt, Frankfurt, Germany, 13 Department of Phoniatrics and Pedaudiology, University Hospital Münster, Münster, Germany

☯ These authors contributed equally to this work.
‡ These authors also contributed equally to this work
* carina.lueke@uni-wuerzburg.de

**Data Availability Statement:** The questionnaires and data from all three survey rounds are available on Open Science Framework: https://doi.org/10. 17605/OSF.IO/6MB2P.

## Abstract

In recent years, there have been intense international discussions about the definition and terminology of language disorders in childhood, such as those sparked by the publications of the CATALISE consortium. To address this ongoing debate, a Delphi study was conducted in German-speaking countries. This study consisted of three survey waves and involved over 400 experts from relevant disciplines. As a result, a far-reaching consensus was achieved on essential definition criteria and terminology, presented in 23 statements. The German term 'Sprachentwicklungsstörung' was endorsed to refer to children with significant deviations from typical language development that can negatively impact social interactions, educational progress, and/or social participation and do not occur together with a potentially contributing impairment. A significant deviation from typical language development was defined as a child's scores in standardized test procedures being ≥ 1.5 SD below the mean for children of the same age. The results of this Delphi study provide a proposal for a uniform use of terminology for language disorders in childhood in German-speaking countries.

**Funding:** This research was funded by the Society for Interdisciplinary Language Acquisition Research and Child Language Disorders in the German-Speaking Countries (GISKID) to the whole D-A-CH Konsortium SES (DACH 2020/1-1; www. giskid.eu). We acknowledge support from the Open Access Publication Fund of the University of Muenster and the University Hospital Münster. CL is the first chair of GISKID and was at the same time intensively involved in the planning and execution of the study and together with CK mainly responsible for the composition of the publication.

**Competing interests:** The authors have declared that no competing interests exist.

## 1. Introduction

Various disciplines and professional groups are concerned with children whose language acquisition is not typical. Over the last decades both, the perspective on atypical language acquisition and the terms to describe and explain the impairments of affected children have significantly changed. So far, there has been a general agreement that language disorders in childhood can occur in the context of other disorders or diseases (e.g., profound developmental disorders or genetic syndromes). The current debate is primarily concerned with children whose language problems cannot be explained (at least in part) by another disorder or impairment. For these children with "unexplained language problems" [1], the term *Specific Language Impairment (SLI)* had been internationally introduced in the 1980s. Following the term SLI, the German synonyms "Umschriebene Sprachentwicklungsstörung (USES) [circumscribed developmental language disorder]" and "Spezifische Sprachentwicklungsstörung (SSES) [specific developmental language disorder]" were established.

Both in English- and German-speaking countries, the terms referring to circumscribed or specific developmental language disorders came under criticism. First, the particular terminology was not applied consistently, but coexisted with a variety of other terms (see [1] for English terms, see [2] for German terms). Moreover, the invoked specificity of specific or circumscribed language disorders was questioned by empirical evidence as well as clinical experience demonstrating that affected children, compared to typically developing children, often show additional delays, impairments or underachievements in other developmental areas like emotional and/or behavioral abilities or auditory processing (for a complete list of potentially affected areas with references see section 3.3.2). This led to a professional debate on the definition of SLI based on exclusion criteria and the use of the attribute "specific", which was summarized in a special issue of the *International Journal of Language and Communication Disorders* in 2014 [3] and resulted in an international acknowledged Delphi study for English-speaking countries [4,5]. Since then, there is an ongoing discussion on the adaptation of the definition criteria of language disorders in childhood and a potential change in terminology in many countries.

The Delphi study ran by a multidisciplinary consortium of 57 professionals (CATALISE) [5] from different English-speaking countries (i.e., primarily the UK, but also Australia, New Zealand, Ireland, Canada, and the USA) led to an adjustment of the definition criteria and terminology of language disorders in childhood: "The term 'language disorder' is proposed for children who are likely to have language problems enduring into middle childhood and beyond, with a significant impact on everyday social interactions or educational progress." [5]. In cases, in which the language disorder occurs in combination with a more complex impairment (differentiating condition) the term "language disorder" is added via the respective, more complex impairment ("Language disorder associated with X)" [5]. Importantly, the term "Developmental language disorder" (DLD) was proposed [5] and has been widely adopted for language disorders occurring without a differentiating condition since then. In addition to the terminology, the definition criteria were also changed. In recognition of recent empirical and clinical evidence the presence of other impairments in cognitive, sensory-motor or behavioral domains no longer precluded the diagnosis of DLD and were accepted as co-occurring conditions.

The changes in the definition and terminology of language disorders in childhood in English-speaking countries were widely reflected in international journals showing that since 2018 an increasing majority of scientific publications on this topic used the term DLD in their title (53% in 2018–94% in 2023) instead of previously SLI (92% in 2012–82% in 2017; based on data from Web of Science). Moreover, the Delphi study in English-speaking countries led to

similar initiatives and changes worldwide. For example, French-speaking countries decided to use the same definition criteria and to integrate the translated English terms (*"Trouble développemental du langage; TDL)"* [6]. In Norway a separate consensus study was conducted, which was based on a translation of the English statements and arrived at comparable results as the Delphi study in English-speaking countries [7]. In addition, there is an ongoing intensive discussion of the definition and terminology of language disorders in childhood in other countries, for example, in Sweden [8] and in Spanish-speaking regions [9].

In German-speaking countries, an in-depth scientific discussion on this topic was initiated in 2018, when it was introduced by a working group at the 10th *Interdisziplinäre Tagung über Sprachentwicklungsstörungen [Interdisciplinary Conference on Developmental Language Disorders] (ISES)*. Subsequently several position papers [10–12] and statements of professional associations and societies, and language experts were published that revealed a wide range of opinions and views on the CATALISE consensus. The *Gesellschaft für interdisziplinäre Spracherwerbsforschung und kindliche Sprachstörungen im deutschsprachigen Raum (GISKID) [Society for Interdisciplinary Language Acquisition Research and Child Language Disorders in the German-speaking Countries]* then took the lead in moderating the discussion and consensus-building process and organized two expert meetings with representatives from as many disciplines, institutions, and German-speaking countries as possible in 2019. The two meetings (with 43 and 33 participants) revealed both strong commitment regarding a revision of the terminology of DLD and clear differences between the involved disciplines. In particular, physicians (who were underrepresented in CATALISE) expressed serious professional concerns about abandoning the term "specific" translated in German as "umschriebene Sprachentwicklungsstörung [circumscribed developmental language disorder]". First, they were not convinced of the necessity to alter the terminology, since the non-exclusivity of circumscribed developmental language disorder had already been recognized in a German interdisciplinary consensus guideline [13]. Second, they were concerned about the compatibility of a modified terminology with the ICD classification. Lastly, they emphasized the importance of distinguishing between childhood language disorders occurring with or without potentially contributing impairments [12]. In contrast, other professional disciplines recognized the need to revise the definition criteria and terminology. This dissent between the disciplines led to the decision to conduct a Delphi study on the definition and terminology of language disorders in childhood in the German-speaking countries. An interdisciplinary and multinational steering committee consisting of the authors of this paper was founded in 2020 under the name *D-A-CH Konsortium SES* (D: Germany, A: Austria, CH: Switzerland, SES: Sprachentwicklungsstörung [developmental language disorder]). The members of this committee, 11 experts on language disorders in childhood, guided the Delphi study in five German-speaking countries from 2020–2022. Two additional experts were involved, specialized in the monitoring and evaluation of Delphi processes in health sciences [14].

Delphi techniques refer to group discussion procedures in which complex topics are evaluated by experts in an iterative and structured process [15,16]. They are based on the assumption that a group of experts with different perspectives on a complex, ambiguous phenomenon associated with a high degree of uncertainty will render a "better" judgment than a single expert would, even if that person was the most knowledgeable one in the field [17]. This does not imply that all surveyed experts will arrive at the same judgment or that a "correct answer" is found [18]. Instead, the aim is a maximum approximation among expert judgments, based on three methodological principles [19–21]:

• Anonymity of the experts: Group effects arising from status or seniority are minimized.

- Selection of experts: As a result of their knowledge and professional interests or concerns, experts are very willing to intellectually deal with a topic in order to give the most well-founded judgment possible.

- Multi-wave surveys: Conducting a Delphi survey in several rounds and sharing the interim results give experts the opportunity to revise their judgments, particularly if they were uncertain in their former judgment. In a new round, respondents can draw upon existing knowledge that may have been neglected or underestimated, take the interim results into account and, if necessary, use additional information sources. Ultimately, the knowledge base is expanded over the course of the Delphi procedure, enabling a better or more reliable assessment of the topic than in a one-time survey.

The *D-A-CH Konsortium SES* developed a questionnaire that was presented to a large group of experts, representing different disciplines dealing with language disorders in childhood across all German-speaking countries.

## 2. Materials and methods

### 2.1 Experts

Experts from all German-speaking countries (Germany, Austria, Switzerland, Luxembourg and Liechtenstein) and all relevant disciplines (speech-language pathology, psychology and neuropsychology, medicine, (clinical) linguistics, speech science, and special needs education) were invited to participate in the Delphi survey. Their expertise covered research, teaching, and/or clinical practice. Potential experts were informed about the Delphi study via email newsletters, at conferences, and through professional associations and were asked to register for participation. The aim was to recruit a sufficient number of experts with respect to the three categories country, professional discipline, and professional activity. As there are currently no charity organizations dedicated to supporting families of children with language disorders in German-speaking countries, we were unable to include participants representing the perspective of these families. The study was carried out in adherence to the guidelines of the German Research Foundation (DFG), the Consortium for the Social, Behavioural, Educational and Economic Sciences (RatSWD) as well as the Declaration of Helsinki. Ethical review and approval were waived for this study since experts were provided with comprehensive written information about the study procedure and aims before being questioned. All participating experts provided informed written consent prior to their involvement in the study.

A total of 875 experts registered, of which 449 participated in the first round of the Delphi process, 352 in the second round, and 308 in the third round. The initial distribution in the three categories was maintained throughout the three Delphi rounds and the subsequent evaluation. In each round, approximately 76% of the respondents were female, with a more balanced gender ratio in medicine, while the majority of participants in the other disciplines were female. In all three rounds, most of the respondents stated that they were currently working in the practical field (around 80%). About 30% of the respondents stated that they were additionally or exclusively active in research and about 40% additionally or exclusively in teaching. Detailed information about the respondents in each round can be found in Fig 1.

### 2.2 Delphi procedure

The *D-A-CH Konsortium SES* conducted a Delphi procedure with three online rounds (Fig 1). The members of the *D-A-CH Konsortium SES* developed the initial questionnaire based on their professional expertise, substantiated by literature reviews. The initial questionnaire

| Process of the Delphi study in German-speaking countries | |
| --- | --- |
| Interdisciplinary steering committee with eleven experts on language disorders in childhood and two methodological experts (*D-A-CH Konsortium SES*) | |
| **Development of the Delphi survey** | |
| Formulation of statements based on professional expertise and current state of research<br>Conducting a pre-test (*n* = 3): Feedback, among others on comprehensibility and manageability of the questionnaire | |
| **Selection of participating experts** | |
| Requested information:<br><br>Country: Germany, Austria, Switzerland, Lichtenstein, Luxembourg<br>Discipline: Speech-language pathology, medicine, special need education, (clinical) linguistics, psychology, neuroscience<br>Area: Research, teaching, and/or clinical practice<br>Expertise: Knowledge of the current scientific discussion | |
| **First Delphi round (March/April 2021)** | |
| *n* = 875 invitations<br>*n* = 449 responses (51%) of:<br>212 speech-language therapists<br>142 physicians<br>40 special need educators<br>31 (clinical) linguists<br>24 psychologists/ neuroscientists | Online survey with:<br>- 8 topic blocks<br>- 9 scenarios<br>- section on personal information and expertise<br>- 33 closed items<br>- 26 open items |
| **Second Delphi round (September/October 2021)** | |
| *n* = 449 invitations<br>*n* = 352 responses (78%) of:<br>172 speech-language therapists<br>103 physicians<br>34 special need educators<br>26 (clinical) linguists<br>17 psychologists/ neuroscientists | Online survey with:<br>- 8 topics blocks (identical to first round)<br>- new block on age ranges<br>- section on personal information and expertise<br>- 27 closed items<br>- 14 open items<br>- feedback on the results of the first round |
| **Third Delphi round (March/April 2022)** | |
| *n* = 352 invitations<br>*n* = 308 responses (87%) of:<br>146 speech-language therapists<br>88 physicians<br>34 special need educators<br>24 (clinical) linguists<br>16 psychologists/ neuroscientists | Online survey with:<br>- 5 topic blocks<br>- section on personal information and expertise<br>- 11 closed items<br>- no open items<br>- feedback on the results of the second round |
| **Evaluation (April 2022)** | |
| *n* = 449 invitations<br>*n* = 179 responses (40%) of:<br>107 speech-language therapists<br>40 physicians<br>18 special need educators<br>7 (clinical) linguists<br>7 psychologists/ neuroscientists | Online survey with:<br>- 46 closed items<br>- questions about motivation, effort, comprehensibility of the survey |

**Fig 1. Flowchart of the Delphi procedure and participants of all three rounds of survey.**

contained eight topic blocks and a short section on the demographic and professional background of the participants. The eight topic blocks focused on:

- Necessity of consensus building across professions within German speaking countries

- Umbrella term for speech, language, and communication difficulties

- Language disorders in childhood (incl. profiles)

  - Language disorders in childhood occurring together with a potentially contributing impairment

  - Language disorders in childhood occurring without a potentially contributing impairment

  - Cognitive abilities in children with language disorders

- Early language difficulties

- Environmentally caused language difficulties

For each block, the *D-A-CH Konsortium SES* first presented the current state of research in information boxes (regarding, for example, linguistic domains, the development of children with DLD in other areas, and the importance of their cognitive abilities). These boxes were followed by standardized items and open questions for additional comments. The standardized items used eight-point Likert scales (1 = "agree completely", 8 = "disagree completely") and nominal scales to ask about preferred terminology. The comment fields were limited to 250 characters in all of the Delphi rounds and, unlike the other standardized questions, were optional. In addition, participants could also indicate on a four-point scale the level of certainty of their response (1 = "absolutely certain", 2 = "rather certain", 3 = "rather uncertain", 4 = "extremely uncertain"). The online survey software Unipark [22] was used for programming and sending out the questionnaire.

Consensus was defined according to the following criteria: For nominal questions, at least 70% of the responses needed to be attributed to one particular term. A cut-off of 70% is typically applied in Delphi surveys and considered to be a moderate consensus criterion [14,23]. In terms of eight-point rating scales, consensus was reached, if the standard deviation of the average responses for this item did not exceed 2.0, meaning that there was a relatively low level of variance among the responses.

Statistical analyses were conducted by two methodological experts utilizing the statistical program R [24]. These two had access to individual participant data during the data collection process, enabling them to link the data sets from the three waves and present each respondent with their own voting behavior from the previous survey round. However, it is important to note that the eleven subject matter experts were never provided with any information that could potentially identify the participants. For all eight-point rating scales were calculated: frequencies, means weighted by the number of experts in the discipline, standard deviation, and significant differences between disciplines using the Kruskal-Wallis test. For the mean values, the mean values per discipline were calculated first, and then the overall mean value was calculated above them. Importantly, during the second and third rounds of the survey, participants were additionally informed about their own previous responses as well as the prior overall results. Moreover, discipline-specific differences were included to ensure transparency. The statistical results were presented as bar charts. The answers given in the comment boxes were analyzed following the principles of the thematic analysis procedure [25] to systematically document the range of arguments for different positions. Arguments from the previous round were divided into pro and contra judgments and presented to the participants.

In the second round, participants received the same questionnaire with standardized and open items as in the first round, but shortened to only those items that had not reached the defined consensus criteria. Questions that appeared to be unclear or ambiguous based on the answers provided were rephrased accordingly. The third round covered the remaining controversial items, some of which were again rephrased to provide more clarity. Items with eight-point scales, open questions and the questions addressing judgment certainty were not included in the third round. For each of the Delphi rounds, participants were given up to four weeks to complete the survey, with a reminder being sent out after three weeks. After completion of the whole Delphi procedure, all participants who had taken part in at least one round received an email from the *D-A-CH Konsortium SES* with a thank-you note and an invitation to evaluate the procedure. The questionnaires and data from all three survey rounds are available on Open Science Framework: https://doi.org/10.17605/OSF.IO/6MB2Pll.

## 3. Results and discussion

Results are reported according to the eight topic blocks, not identical to the original order in the questionnaires. The numbered statements (1–23) reflect the final set of results, representing the statements and terms that have met the previously established consensus criteria. In the Delphi questionnaire, these statements were presented in German and are here presented in their English translation. Note that the terminology is given in both German and its English translation. These English translations were of course not voted in the present Delphi study and would possibly be chosen differently by other persons.

### 3.1 Necessity of consensus building

1. It is useful that all persons working professionally with language difficulties in children and adolescents use uniform terminology to describe language impairments.

2. A uniform definition and terminology should be used in all German-speaking countries.

3. A uniform definition and terminology should be supported by all relevant professional groups.

For all of these three statements, a consensus was reached with very high approval ratings ($M_1$ = 1.4, $SD_1$ = 1.0, $M_2$ = 1.2, $SD_2$ = 0.7, $M_3$ = 1.8, $SD_3$ = 1.6). This shows that there is a very high agreement among all the professional groups involved from all German-speaking countries about establishing uniform definition criteria for childhood language disorders and using uniform terminology.

### 3.2 Umbrella term for speech, language, and communication difficulties

- In childhood and adolescence, different areas of speech, language, voice, speech fluency, and communication can be affected. Therefore, it is useful to define an umbrella term, under which different forms of difficulties in the mentioned areas fall.

4. The umbrella term, under which different forms of difficulties in the mentioned areas fall, is: "Sprach-, Sprech- und Kommunikationsauffälligkeiten [Language, speech and communication difficulties]".

5. This term is specified by the supplement "im Kindes- und Jugendalter [in childhood and adolescence]".

The need of defining an umbrella term for speech, language, and communication difficulties was rated quite heterogeneously by participants ($M$ = 3.3, $SD$ = 2.1), so that no consensus was reached. However, if there were such an umbrella term, 89% agreed that it should be called "Sprach-, Sprech- und Kommunikationsauffälligkeiten [Language, speech and communication difficulties]". Furthermore, this term should be supplemented by the supplement of "im Kindes- und Jugendalter [in childhood and adolescence]" according to 82% of the participants. Therefore, the term "Sprach-, Sprech- und Kommunikationsauffälligkeiten im Kindes- und Jugendalter [Language, speech and communication difficulties in childhood and adolescence]" was established as an umbrella term for all difficulties of speech, language, voice, fluency and communication in children and adolescents, whereby the actual need for such an umbrella term will be shown by the future application practice. Importantly, these difficulties are not necessarily classified as disorders in a clinical sense. This also means that the assignment of a child or adolescent to difficulties under this umbrella term does not necessarily imply an indication for therapy. In each individual case, the appropriate form of support will have to be chosen on the basis of a comprehensive diagnosis.

### 3.3 Language disorders in childhood

6. Childhood language disorders are defined as significant deviations from typical language development that can negatively impact children's social interactions, educational progress, and/or social participation. These disorders may co-occur with a potentially contributing impairment or independently of such an impairment.

7. These childhood language disorders are called "Sprachentwicklungsstörungen (SES) [developmental language disorders (DLD)]".

8. A significant deviation from typical language development is determined by standardized tests in addition to observation and interview procedures. A significant deviation is determined, if the child's test scores are at least 1.5 standard deviations below the mean of children at the same age. This corresponds to a T score of < 35 or a percentile rank of < 7.

For the 6th statement, a consensus was reached with high agreement ($M$ = 1.7, $SD$ = 1.4), thus resulting in a differentiation between language disorders occurring together with a potentially contributing impairment and language disorders without the presence of such a disorder. With a response of 78%, there was a consensus to use the term "Sprachentwicklungsstörung [developmental language disorder]" to describe these language disorders in childhood. A consensus was also reached on the definition criteria of a significant deviation from typical language development, as determined by standardized tests in addition to observation and interview procedures. Confirmed by 79% of the participants, a significant deviation is assumed if a child's test scores in a standardized test procedure are $\geq$ 1.5 $SD$ below the mean of children at the same age.

9. Multilingual children can only be classified as having a developmental language disorder if they do not demonstrate age-appropriate abilities in any of their spoken languages.

Participants agreed on this statement with a mean response of 2.4 ($SD$ = 2.0). This means that multilingual children cannot be classified as having a developmental language disorder if they perform age-appropriate in any of their spoken languages. This agreement acknowledges that children growing up with two or more languages should not exclusively be tested in only one of their languages in order to determine the presence of a developmental language disorder. It is well documented that multilingual, typically developing children often perform below

average in standardized tests compared to monolingual children, especially regarding their vocabulary [26]. Therefore, using monolingual norms for multilingual children leads to a significantly increased rate of false positives in diagnosing developmental language disorders [27]. As a consequence, a theoretically and empirically driven adaptation of diagnostic procedures for multilingual children is necessary, in which the input factors of the respective languages are considered, and children's linguistic competencies are adequately assessed in all of their acquired languages. Due to the heterogeneity of multilingual acquisition trajectories and the fact that there are not sufficient test procedures for developmental language disorders in multilingual children, the listed cut-off value of 1.5 *SD* below the mean as a diagnostic criterion cannot be applied to multilingual children. Moreover, children with their home language differing from the environmental language (with insufficient exposure to the latter) might have language support needs, but not necessarily a developmental language disorder.

10. Children with developmental language disorders show heterogeneous profiles of individual strengths and weaknesses, with receptive and/or expressive deficits at one, several, or all linguistic domains.

11. The individual symptoms of children with developmental language disorders can be described and specified on the basis of the different linguistic domains. However, the respective profiles do not constitute clearly definable subgroups.

12. Phonological disorders are both, a component of developmental language disorders and a form of speech sound disorders.

These three statements regarding the particular symptoms and profiles of children with developmental language disorders were confirmed by a consensus with very high agreement ratings ($M_{10}$ = 1.1, $SD_{10}$ = 0.5, $M_{11}$ = 1.7, $SD_{11}$ = 1.3, $M_{12}$ = 2.2, $SD_{12}$ = 1.8). Thus, the description of receptive and expressive symptoms at the different linguistic domains will enable the identification of different profiles of disorders. Deficits can occur in the following linguistic-communicative domains:

- Phonology: The ability to select and combine speech sounds (phonemes) to build and organize sound sequences and word forms.

- Lexicon and semantics: The ability to establish and organize word meanings, to build an adequate vocabulary of different parts of speech, and to access or retrieve words from the lexicon (word finding).

- Morphology and syntax: The ability to use the grammatical rules of a given language to form words, sentences, and texts.

- Pragmatics and social communication: The ability to understand/express utterances in which the intended meaning goes beyond what is said (figurative language), construct conversations and discourse, build narratives coherently in terms of content and form, as well as use verbal and nonverbal communicative means adapted to the situation.

### 3.3.1 Developmental language disorders occurring together with a potentially contributing impairment.

13. Developmental language disorders may co-occur together with another, more complex impairment. This impairment may causally contribute to the language disorder.

14. Possible impairments that may contribute to developmental language disorders are:

- intellectual disabilities (non-verbal IQ below 70)

- hearing disorders

- genetic syndromes

- autism spectrum disorders

- neurodegenerative diseases

- infantile brain damage

- childhood aphasia

- motor disorders

With particularly high levels of agreement ($M = 1.3$, $SD = 0.9$), the 13[th] statement reached consensus. Participants rated the listed impairments as possible contributions to a developmental language disorder with 96% agreement for intellectual disabilities, 95% for hearing disorders, 94% each for genetic syndromes and autism spectrum disorders, 93% for infantile brain damage, 91% for neurodegenerative diseases, 79% for childhood aphasia, and 73% for motor disorders.

15. These disorders are called "Sprachentwicklungsstörungen bei X" or "Sprachentwicklungsstörung assoziiert mit X [in English both: developmental language disorders associated with X]"

16. The term developmental language disorder associated with X can be used for all children and adolescents with no defined lower or upper age limits.

To refer to children with a developmental language disorder occurring together with a potentially contributing impairment, two additional expressions to the term "Sprachentwicklungsstörung (SES) [developmental language disorder]" were chosen at equal parts: "bei X" (46%) or "assoziiert mit X" (43%) ["with X" and "associated with X"]. Consequently, this leads to expressions like "SES bei Down Syndrom/SES assoziiert mit Down-Syndrom [developmental language disorder with/associated with Down syndrome]". The designation of childhood aphasia is the only exception to this. While childhood aphasia does interfere with language development, it is not a developmental language disorder, but an acquired language disorder. The inclusion of childhood aphasia in the list of potentially contributing factors prevents neglect or lack of awareness of this disorder. In this specific case, however, the term "childhood aphasia" is sufficient and there is no need to couple it with the term "Sprachentwicklungsstörung [developmental language disorder]".

The majority of participants rated against the application of an upper age limit (64%) for using these descriptions. Among those participants, who indicated an upper age limit for the application of the term "SES assoziiert mit X [developmental language disorder associated with X]", 10% indicated the end of or beyond adolescence (values $\geq 18$ years). Based on the fact, that 74% of the participants voted against an upper age limit or suggested it beyond adolescence, no upper age limit for this group was defined. Similarly, the majority of participants (65%) were opposed to differentiating between young and older children and thus they did not support the implementation of a lower age limit. Participants who favored a lower age limit, gave values with a median of 12 months. From these combined results, it follows that the term "SES assoziiert mit / bei X [developmental language disorder with X/ associated with X]" can be used uniformly for all children and adolescents with significant deviations from typical language development that can negatively impact children's social interactions, educational

progress, and/or social participation and occur together with a potentially contributing impairment.

**3.3.2 Developmental language disorders occurring without a potentially contributing impairment.**

17. Developmental language disorders can occur without a potentially contributing impairment.

18. The definition criteria for developmental language disorders occurring without a contributing impairment need to be updated to reflect the current state of research.

With high levels of agreement, a consensus was reached for the 17th ($M$ = 1.4, $SD$ = 0.7) and 18th ($M$ = 1.8, $SD$ = 1.5) statements. Prior to rating the 18th statement, participants were informed about the current state of research regarding co-occurring difficulties in several domains whose causal relation to language difficulties is yet unclear. Specifically, the following information was provided:

"A large number of studies [28] have shown that developmental language disorders that occur without a potentially contributing impairment can be accompanied by other disorders or underachievements. These disorders and underachievements may include:

- impairments in auditory perception and processing [29–33]

- deficits in executive functions (i.e., cognitive functions that control behavior and experience, usually including working memory, response inhibition, and cognitive flexibility as subcomponents) [34–38]

- deficits in phonological working memory [39–42]

- mild abnormalities in gross, fine, or speech motor skills [43,44] or motor development disorders, orofacial disorders

- social and emotional difficulties or behavioral problems [45], increased risk for internalizing (e.g., anxiety and depression [46–48]) and externalizing disorders (e.g., ADHD [49–51])

- developmental dyslexia [52–54]."

19. The term for language disorders in childhood that occur without a potentially contributing impairment is "Sprachentwicklungsstörung [developmental language disorder]".

20. The term developmental language disorder can be used for all children and adolescents from the age of 3;0 years.

Referring to children with a developmental language disorder occurring without a potentially contributing impairment, 67% voted against an additional supplement like "umschrieben [circumscribed]" or "spezifisch [specific]" of the term "Sprachentwicklungsstörung (SES) [developmental language disorder]". Although the consensus criterion for nominal items (agreement ≥ 70%) was thus not reached, the D-A-CH Konsortium SES decided to follow the opinion of the majority of the participants and therefore recommends using the term "Sprachentwicklungsstörung (SES) [developmental language disorder]" without any additional specification.

With respect to the age range, for which the term "Sprachentwicklungsstörung (SES) [developmental language disorder]" should be used, 72% of participants rated for defining a lower age limit, with a median age of 36 months. In contrast, more than half of the participants (59%) rated against setting an upper age limit. Those participants who indicated an upper age limit, this was set with 24% at the end of or beyond adolescence (values ≥ 18 years). Given that

83% of the participants voted against an upper age limit or suggested it to be set beyond adolescence, no upper age limit was defined. Therefore, the term "Sprachentwicklungsstörung (SES) [developmental language disorder]" is recommended, if these disorders do not occur together with a potentially contributing impairment, for children and adolescents at $\geq$ 3;0 years of age, who experience significant deviations from typical language development that can negatively impact children's social interactions, educational progress, and/or social participation.

**3.3.3 Cognitive abilities in children with language disorders.** For the diagnosis of developmental language disorders, there has always been a debate about the role of nonverbal cognitive abilities. First, it is generally acknowledged that developmental language disorders can occur at average or above-average nonverbal intelligence (IQ scores of > 85). Second, there is common agreement to consider severely impaired non-verbal cognitive abilities (IQ score < 2 SD below the mean, i.e., < 70) as a co-causing condition of developmental language disorders. However, it is controversial whether less substantial difficulties in nonverbal intelligence (IQ scores of 70–85, i.e., between 1–2 SD below the mean) should also be classified as a potentially co-causing factor in developmental language disorders or not. This controversy was addressed with two questions in the Delphi-survey:

- In which category should children with developmental language disorders with nonverbal IQ scores slightly below average (85–70) be placed?

- At what IQ score would you place a child with low nonverbal cognitive skills in the category of "developmental language disorders occurring together with a potentially contributing impairment"?

In the first question, 48% of the participants categorized children with a nonverbal IQ score between 85–70 as belonging to the group of children with "developmental language disorders associated with X", while only 23% of participants assigned those children to the group "developmental language disorders". Yet, 60.5% of the same participants also defined a nonverbal IQ score of < 70 (and not < 85) as a potentially contributing impairment as an answer to the second question. As these ratings indicate, no consensus regarding the role of nonverbal cognitive abilities in developmental language disorders was reached. Moreover, based on the contradictory nature of the current results, no recommendation for the consideration of defined IQ-score ranges of nonverbal cognitive abilities in developmental language disorders was given by the *D-A-CH Konsortium SES*. Notwithstanding, it remains important to emphasize that the overall development of children should be evaluated, including an assessment of their cognitive abilities.

## 3.4 Early language difficulties

Language and communication difficulties can occur early during childhood, i.e., within the child's first three years of life. Participants had already agreed that no distinction should be made between younger and older children with a potentially contributing impairment (see statement 16). As a result, even young children, under three years of age, can be referred to as having a developmental language disorder associated with X, if their language skills are significantly below the age-typical language status.

21. Children without a potentially contributing impairment may show difficulties in language and communication development from early on (within the first three years of life). A specific category is appropriate to describe these cases.

22. For these young children, the terms "late talker" or "Sprachentwicklungsverzögerung [developmental language delay]" can be used synonymously.

Participants agreed on statement 21 ($M$ = 2.1, $SD$ = 1.9), which is in line with the definition of a lower age limit for children with developmental language disorder without a potentially contributing impairment at the age of 3;0 years. Yet, no consensus was reached as to what term should describe the group of children named in statement 21. Specifically, the two terms "late talker" and "Sprachentwicklungsverzögerung [developmental language delay]", which are both commonly used in German-speaking countries, were chosen with almost equal proportions (i.e., 51% and 49%, respectively). Therefore, these two terms can be used synonymously to refer to children younger than three years of age with significant deviations from typical language development, but without a potentially contributing impairment.

### 3.5 Environmentally caused language difficulties

- Language difficulties with comparable characteristics as those in developmental language disorders can also be caused environmentally by to little or qualitatively deficient language input.

1. If language difficulties are caused by insufficient language input, the term "Umgebungsbedingte Sprachauffälligkeit [environmental language difficulty]" should be used.

- This category also includes monolingual children growing up in a socially disadvantaged environment. The category moreover includes children growing up multilingually, who show age-appropriate skills in their language spoken at home, but non-appropriate skills in their ambient language, caused by a lack of familiarity with the ambient language.

Participants' responses revealed a disagreement on the necessity for a category to describe children with a need for language support due to environmental factors. Participants showed some agreement on the first item with a mean response of 3.2, but due to the large response variability of 2.4 $SD$, no consensus was reached. With respect to labelling such a category, however, the term "Umgebungsbedingte Sprachauffälligkeit [environmental language difficulty]" was chosen by 73% of the participants. This term has already been established by the German guidelines for the diagnosis of developmental language disorders [13] and was thus confirmed by the current results. The third item to be rated, which describes the group of children with environmental language difficulties in more detail, again resulted in some agreement, but no consensus was reached due to a very large variability in responses ($M$ = 3.4, $SD$ = 2.7). Note that this disagreement likely results from suboptimal phrasing of the item, including two different groups in this category, rendering a response ambiguous. Thus, a lack of agreement on this statement might be driven by concerns about including either monolingual children from a socially disadvantaged background or multilingual children with insufficient input in the environmental language in this category. In the English-language classification [5], a similar category called "lack of familiarity with ambient language" is used for multilingual children only.

### 4. General discussion

The current study aimed at clarifying the definition criteria of language disorders in childhood and resolving conceptual ambiguities of terminology used in German-speaking countries, following similar initiatives in English-speaking countries [4,5] and Norway [7]. To this end, the *D-A-CH Konsortium SES* was founded to run a Delphi procedure on the definition criteria and terminology in German-speaking countries with more than 400 participants over a period of

two years. Further explanations and a diagram of the German terms are presented by Kauschke et al. [2].

As a key result of the current Delphi study on language disorders in childhood, the *D-A-CH Konsortium SES* recommends by majority (67%) leaving out a restrictive specifier, such as "spezifisch [specific]" or "umschrieben [circumscribed]", when referring to children with significant deviations from typical language development that can negatively impact children's social interactions, educational progress, and/or social participation and do not occur together with a potentially contributing impairment. The lack of consensus (67% instead of 70% agreement) can be attributed to the difference between the medical profession and the other disciplines. Among physicians, 59% voted in favor of retaining a specifier. Here, the specifier "circumscribed" has been mainly used in medical contexts, while "specific" has been predominantly used in speech-language pathology and therapy. Physicians likely considered the use of such a specifier as essential to their daily work, i.e., a diagnostic differentiation between children with and without potentially contributing impairments, acknowledging that the latter may still experience mild difficulties in other areas (see 18[th] statement). The position of the medical disciplines is further discussed in Neumann et al. [12].

Previously, the term "Sprachentwicklungsstörung [developmental language disorder]" had been used as a superordinate term for language disorders in children with and without a potentially contributing impairment, and this was also confirmed in this Delphi study (6[th] and 7[th] statement). Thus, in German, unlike in English, this term includes children without a potentially contributing impairment, but also children with such an impairment. In the latter case, however, the further impairment must be explicitly named (13[th]-15[th] statements). This implies that the solely use of the term "Sprachentwicklungsstörung [developmental language disorder]" now exclusively describes children without a potentially contributing impairment, similar to the English term *Development Language Disorder* (DLD) (see [1,7]).

Interestingly, participants did not agree on the necessity of an umbrella term that covers all different forms of difficulties in the areas of speech, language, voice, fluency and communication. For those who wish to use an umbrella term, it was agreed to be "Sprach-, Sprech- und Kommunikationsauffälligkeiten im Kindes- und Jugendalter [Language, speech and communication difficulties in childhood and adolescence]", which is comparable to the English concept of "Speech, Language and Communication Needs (SLCN)" [5]. However, its use in German-speaking countries seems less uniform and consistent.

Specifying the definition criteria for developmental language disorders, the current Delphi study revealed a consensus for the statement that children with developmental language disorders show heterogeneous profiles of individual strengths and weaknesses, with receptive and/or expressive deficits at one, several, or all linguistic domains. Participants moreover agreed that the individual profiles of language disorders can be further described and specified on the basis of the different linguistic domains, while these profiles do not constitute clearly definable subgroups.

In addition, consensus was reached for the case that children with developmental language disorders may also show delays or impairments in other developmental areas whose causal relation to the linguistic deficits are unclear, with this not precluding the diagnosis of developmental language disorder. In contrast, developmental language disorders in childhood can also occur together with a potentially contributing impairment. Here, the majority of participants was in favor of making these cases explicit by using either one of the two supplements: "Sprachentwicklungsstörungen bei X/Sprachentwicklungsstörung assoziiert mit X [developmental language disorder with X/associated with X]", which is comparable to the term "Language disorder associated with X", established in English-speaking countries [5]. Undoubtedly, a thorough and comprehensive assessment of all developmental domains in

children with language development disorders is crucial for identifying any concurrent or associated disorders and for initiating tailored interventions as needed.

In the context of factors contributing to developmental language disorders in childhood, the role of cognitive abilities, specifically mild cognitive impairments (i.e., IQ between 85 and 70), could not be clarified in the current Delphi study. No consensus was reached regarding the classification of mild cognitive impairments as a co-occurring condition (i.e., category of *developmental language disorders*) or as factor contributing to language disorders in childhood (i.e., category of *developmental language disorders associated with X*). This null result stands in contrast to the CATALISE outcome, for which low cognitive abilities do not preclude a diagnosis of developmental language disorder [5]. In addition, reconciling the issue of cognitive functioning with the ICD-11 [55], poses a challenge: For the superordinate category developmental speech or language disorders (6A01) the ICD-11 states: *"Developmental speech or language disorders arise during the developmental period and are characterized by difficulties in understanding or producing speech and language or in using language in context for the purposes of communication that are outside the limits of normal variation expected for age and level of intellectual functioning."* [55]. However, there is no specific threshold given clarifying what is meant by *"outside the limits of normal variation"*. For the subordinate category of developmental language disorder (6A01.2) the ICD-11 states: „*The individual's ability to understand, produce or use language is markedly below what would be expected given the individual's age. The language deficits are not explained by another neurodevelopmental disorder or a sensory impairment or neurological condition, including the effects of brain injury or infection.*" [55]. While cognitive abilities are deemed relevant for the superordinate category, they are not included in the definition of the more specific subcategory DLD. As the present Delphi study did not reveal a clear result regarding the role of cognitive abilities in children with language disorders, further clarification and discussion are required. From a medical perspective, Neumann et al. [12] has suggested a modified cognitive criterion and emphasized the significance of identifying mild nonverbal deficits as a crucial part of the diagnostic process.

As the learning environment impacts on a child's language development, multilingual children can only be classified as having a developmental language disorder if they do not demonstrate age-appropriate abilities in any of the acquired languages. Here, the current Delphi study revealed consensus that language difficulties resulting from an insufficient exposure to the ambient language are not considered as developmental language disorder, but rather as environmental language difficulties (see also, German guidelines for the diagnosis of developmental language disorders [13]). Although these questions were included in the current Delphi study, the thorough assessment of language capacities of multilingual children will require more comprehensive studies in international and multilingual contexts.

When comparing the overall results of the current study with the outcome of the Delphi study in English-speaking countries [4,5], there are striking parallels regarding the definition criteria of language disorders in childhood, also reflected in the respective terminology. Yet, there are also relevant differences (summarized in Table 1), such as the role of cognitive abilities in the classification of developmental language disorders, as discussed above. Moreover, the *D-A-CH Konsortium SES*, unlike the *CATALISE consortium* of the Delphi study in English-speaking countries [5], does not include the prognosis of a child's language development as a defining characteristic of developmental language disorders. This decision is based on the challenge to identify reliable criteria for evaluating the progression of developmental language disorders and to determine the cases expected as "unlikely to resolve without specialist help" [5]. Instead, the *D-A-CH Konsortium SES* focuses on clarifying the meaning of *significant deviation from typical language development* in the definition of developmental language disorders. Participants agreed that a significant deviation from typical language development is given when a

**Table 1. Divergent results of CATALISE [5] and D-A-CH Konsortium SES.**

| Component | CATALISE | D-A-CH Konsortium SES |
|---|---|---|
| Umbrella term for speech, language, and communication difficulties | Speech, Language, and Communication Needs (SLCN) | German term: "Sprach-, Sprech- und Kommunikationsauffälligkeiten"[language, speech, and communication difficulties]<br>→ Necessity of such an umbrella term open to discussion |
| Language disorders in childhood–Key diagnostic criteria | Prognosis | Significant deviation from typical language development: the child's test scores are at least 1.5 standard deviations below the mean of children at the same age. |
| Term DLD and age range | "The term Developmental Language Disorder (DLD) is proposed to refer to cases of language disorder with no known differentiating condition" ([5] p. 1071)<br>→ No age range was specified | Corresponding German term: "Sprachentwicklungsstörung (SES)"[developmental language disorder]<br>→ Age range: the term SES applies to all children and adolescents from the age of 3;0 years. |
| Phonological disorders | Preschoolers with phonological disorders only do not meet the criteria for DLD. A diagnosis of DLD or a dual diagnosis of DLD with speech sound disorder (SSD) is not appropriate before the age of five ([5] p. 1073). | Regardless of age, phonological disorders are considered as both, a component of DLD and a form of SSD. |
| Cognitive abilities in children with DLD | "A child with a language disorder may have a low level of nonverbal ability. This does not preclude a diagnosis of DLD." ([5] p. 1072) | No consensus was reached about the role of nonverbal cognitive abilities for the assignment of children to a category (developmental language disorder or developmental language disorder associated with X). |
| Early language difficulties | Not part of the study. | Children below the age of 3 years, showing language difficulties in the absence of associated disorders are labelled as "late talkers" [English term similarly used in German] or "Kinder mit Sprachentwicklungsverzögerung" [children with developmental language delay] |
| Unfamiliarity with ambient language | "Some children may have language needs because their first or home language differs from the local language, and they have had insufficient exposure to the language used by the school or community to be fully fluent in it." ([5] p. 1071) | If language difficulties are caused by insufficient language exposure, the German term "Umgebungsbedingte Sprachauffälligkeit" [environmental language difficulty] was consented, but<br>→ there was no full consensus on the definition of the category. It remains open, which children should be included in this group. |

child's scores in standardized tests are 1.5 *SD* or more below the mean when compared to children of the same age. This consensus, when supported across disciplines, may lead to a standardization of diagnostic practice in clinical and educational settings and contribute to better comparability of empirical evidence. It should be noted, however, that the criterion of tests scores below 1.5 *SD* can only be implemented if adequately standardized tests are available. A compilation of standardized German tests that assess a range of language skills in various domains can be found in Spreer [56]. However, the majority of these tests are designed for standard German. This means that the tests may not be fully applicable to Swiss German, Austrian German, or other regional varieties of German. Generally, this criterion of 1.5 *SD* below the mean only applies to children growing up monolingually, while the assessment of multilingual children requires a theoretically and empirically based adaptation of the tests designed and standardized for monolingual children [27,57]. In addition to the results of standardized tests, a diagnosis of developmental language disorder should ultimately be accompanied by observation and interview procedures.

In our Delphi study, we also aimed at specifying the age ranges for which the defined terms describing language difficulties in children apply. For children with a developmental language disorder associated with a potentially contributing impairment, participants voted for the label "Sprachentwicklungsstörung bei X/Sprachentwicklungsstörung assoziiert mit X [developmental language disorder with X/associated with X]" throughout childhood and adolescence (i.e., without a lower or upper age limit). For children with a developmental language disorder without a potentially contributing impairment, however, the term *disorder* is used only from the age of 3;0 years. Here, participants agreed on referring to language and communication difficulties occurring before the age of 3;0 by either of the terms "Late talker" or

"Sprachentwicklungsverzögerung [developmental language delay]". Although the label *disorder* is not used during this limited age period, it is acknowledged that an early delay is associated with a significantly increased risk of developmental language disorders [58]. Early delays should thus be closely monitored, and appropriate measures taken in due time. Here, parent- and child-centered early interventions have been shown to be effective [59–61].

Future research will have to extend the current work by more closely studying developmental language disorders in multilingual children as well as further evaluating the role of cognitive abilities. Independent of the classification of cognitive abilities as co-occurring or contributing conditions to language disorders, all children with diagnosed developmental language disorders need to receive therapy tailored to their nonverbal cognitive performance levels. Additionally, complementary intervention approaches might be appropriate in individual cases addressing nonverbal cognitive deficits.

The current Delphi study has several methodological strengths, given the highly interdisciplinary steering group and the large number of participating experts throughout three survey rounds, representing all relevant professional fields in five German-speaking countries. Despite these advantages, there were also methodological limitations that need to be considered when interpreting the outcome of the current study and when designing future Delphi procedures on the topic of language disorders in childhood. First, our consensus criterion of 70% for nominal questions was lower than the criterion of 75% used in the Delphi study in English-speaking countries [5]. As there is currently no generally defined consensus criterion [14,23], we set the moderate criterion of 70%, which seems justified given the large number of participants in our study. For a comparison, changing the consensus criterion to 75% in the final Delphi round, only the assessment of motor disorders as a contributing condition of developmental language disorders and the ratings on the lower age limits would have not led to a consensus, while all other results remained the same.

Second, the selection of experts participating in the Delphi study was largely unsupervised. Thus, although the call for study participation was sent to all relevant disciplines in all German-speaking regions, it cannot be ruled out that this call was more strongly advertised and supported by individual persons or organizations than by others. This could have led to biases in the proportion of people who participated from different disciplines or regions. We partly addressed potential biases by first calculating the mean values of the rating scales for each of the five disciplines separately, and then computing their mean. Moreover, while it can be assumed that the degree of expertise on the topic varied within the participant sample, this might have been partly accounted for by the large number of participants (i.e., >400).

Third, in a Delphi study, the precise, neutral, and unambiguous phrasing of items allows for representative results of consensus or disagreement. Yet, in our study, the wording on environmental language difficulties was ambiguous, which most likely led to the lack of consensus. Independent of this shortcoming, the definition and identification of developmental language disorders in multilingual children requires further research and an international and multilingual Delphi study.

## 5. Conclusion

The present comprehensive Delphi study with more than 400 experts in speech-language pathology, psychology and neuropsychology, medicine, (clinical) linguistics, speech science and special needs education from Germany, Austria, Switzerland, Liechtenstein, and Luxembourg delivers clear and valuable recommendations regarding the definition and terminology of language disorders in childhood. In the areas for which no consensus was reached, further agreement processes seem necessary. As a consequence, a more unified understanding of

language disorders and the joint use of the proposed terms will promote interdisciplinary exchange and research, facilitate communication with affected families, and overall increase awareness of this disorder among the general population and research funders.

## Acknowledgments

We thank all experts for their participation and Kerstin Schröter for her support in the (qualitative) data analysis.

## Author Contributions

**Conceptualization:** Carina Lüke, Christina Kauschke, Andrea Dohmen, Andrea Haid, Christina Leitinger, Claudia Männel, Tanja Penz, Steffi Sachse, Wiebke Scharff Rethfeldt, Julia Spranger, Susanne Vogt, Marlen Niederberger, Katrin Neumann.

**Data curation:** Julia Spranger, Marlen Niederberger.

**Formal analysis:** Julia Spranger, Marlen Niederberger.

**Funding acquisition:** Carina Lüke, Christina Kauschke, Katrin Neumann.

**Project administration:** Carina Lüke, Marlen Niederberger.

**Resources:** Carina Lüke, Christina Kauschke, Andrea Dohmen, Andrea Haid, Christina Leitinger, Claudia Männel, Tanja Penz, Steffi Sachse, Wiebke Scharff Rethfeldt, Julia Spranger, Susanne Vogt, Marlen Niederberger, Katrin Neumann.

**Visualization:** Christina Kauschke.

**Writing – original draft:** Carina Lüke, Christina Kauschke, Marlen Niederberger.

**Writing – review & editing:** Andrea Dohmen, Andrea Haid, Christina Leitinger, Claudia Männel, Tanja Penz, Steffi Sachse, Wiebke Scharff Rethfeldt, Julia Spranger, Susanne Vogt, Katrin Neumann.

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
