## [Decision Letter · Decision Letter 0]

19 Sep 2023

PONE-D-23-16829Definition and terminology of Developmental Language Disorders - Interdisciplinary consensus across German-speaking countriesPLOS ONE

Dear Dr. Lüke,

Thank you for submitting your manuscript to PLOS ONE. After careful consideration, we feel that it has merit but does not fully meet PLOS ONE’s publication criteria as it currently stands. Therefore, we invite you to submit a revised version of the manuscript that addresses the points raised during the review process.

**The three Reviewers have really appreciated the manuscript and recommended its publication. However, two of them, in particular, provided suggestions and comments for its improvement to be submitted to the authors before publication. Therefore, I urge the authors to accept and address the Reviewers’ comments by proceeding to revise the manuscript accordingly.**

We look forward to receiving your revised manuscript.

Kind regards,

Stefano Federici, Ph.D.

Academic Editor

PLOS ONE

3. We noted in your submission details that a portion of your manuscript may have been presented or published elsewhere. [A German article on the Delphi study will be published in Logos for the use of German-speaking practitioners, while the current manuscript is aimed at the international scientific community.] Please clarify whether this publication was peer-reviewed and formally published. If this work was previously peer-reviewed and published, in the cover letter please provide the reason that this work does not constitute dual publication and should be included in the current manuscript.

Additional Editor Comments:

The three Reviewers have really appreciated the manuscript and recommended its publication. However, two of them, in particular, provided suggestions and comments for its improvement to be submitted to the authors before publication. Therefore, I urge the authors to accept and address the Reviewers’ comments by proceeding to revise the manuscript accordingly.

Reviewers' comments:

Reviewer's Responses to Questions

**Comments to the Author**

1. Is the manuscript technically sound, and do the data support the conclusions?

Reviewer #1: Yes

Reviewer #2: Yes

Reviewer #3: Yes

2. Has the statistical analysis been performed appropriately and rigorously? 

Reviewer #1: Yes

Reviewer #2: Yes

Reviewer #3: N/A

3. Have the authors made all data underlying the findings in their manuscript fully available?

Reviewer #1: Yes

Reviewer #2: No

Reviewer #3: Yes

4. Is the manuscript presented in an intelligible fashion and written in standard English?

Reviewer #1: Yes

Reviewer #2: Yes

Reviewer #3: Yes

5. Review Comments to the Author

Reviewer #1: The authors report on the results of a delphi study on the definition and terminology of language disorders in childhood.

This is an important topic and the authors report convincingly the history and background of this endeavor. The experts involved represent all relevant professional fields. Results are clearly presented and discussed. It is also clearly stated when no consensus could be reached. Strengths and limitations are well discussed.

Two thoughts:

- Statement 9. Multilingual children can only be classified as having a developmental language disorder if they do not demonstrate age-appropriate abilities in any of their spoken languages. This sentence is difficult to understand, even with the paragraph explaining it. Could the authors add a sentence such as ‘this means that multilingual children cannot be classified as having a developmental language disorder if they are age-appropriate in one of their spoken languages’ –following ‘Participants agreed on this statement with a mean response of 2.4 (SD = 2.0).’

- The aspect of difficulties in nonverbal intelligence together with a developmental language disorder is of high importance. It is interesting to read that no consensus regarding the role of nonverbal cognitive abilities in developmental language disorders (whether a potentially co-causing factor) was reached when considering IQ-score ranges and, thus, no recommendation for the consideration of defined IQ-score ranges of nonverbal cognitive abilities in developmental language disorders was given. But could the authors add a sentence that an evaluation of cognitive abilities is of high importance in a child with language delay – as is also discussed later (lines 665-666).

Reviewer #2: It’s perhaps not surprising that I’ve got very few comments, given that this is a report of a Delphi project that was carefully planned and co-ordinated with many co-authors, and the paper is a straightforward account of the final statements.

The aim of the study was to use the Delphi method to create an agreed definition and terminology for childhood language disorders in German-speaking languages. It is an excellent idea to publish an English translation of the outcome, to facilitate comparisons across languages. This is a major study that should have real impact in streamlining research in German-speaking countries and facilitating comparisons with English-speaking groups.

As with the CATALISE English-language study, a wide range of professional groups was included, ranging from those with medical, educational and linguistics expertise.

The one limitation, compared to CATALISE, was the lack of any representation from charities working with language-impaired groups – I suspect these may not exist in the German-speaking countries. In general, it can be good to include the ‘user perspective’, but it is admittedly difficult when the condition under consideration is language disorder, and the discussions are all about which words to use. But this would be worth a mention.

I could see no mention of open data. It would be useful to make the anonymised reports from the panel openly available; I suspect these won’t attract much interest currently, but when the history of children’s language disorders comes to be written, such information will be invaluable.

It was good to see broad agreement between this German-language Delphi and the English-language CATALISE. It might be worth saying a little more about the recommendation re using standardized tests to define disorder. This, of course, has the limitation that it requires adequately standardized tests, which may not be available in all countries. I wondered if it might be worth adding an Appendix with a few guidelines regarding suitable standardized tests?

I think it would be helpful to have a little table at the end that lists the points where the D-A-CH SES consortium and CATALISE consortium diverge. This could help set an agenda for future research that might bring different language definitions into closer alignment.

Finally, it might be worth inventing an acronym to make it easy to refer to this study!

**Minor wording suggestions**

87 dispute -> debate

88-89: the journal is the International Journal of Language and Communication Disorders, and the summary article would be worth citing here:

Reilly, S., Bishop, D. V. M., & Tomblin, B. (2014). Terminological debate over language impairment in children: Forward movement and sticking points. International Journal of Language & Communication Disorders, 49(4), 452–462. https://doi.org/10.1111/1460-6984.12111

101-102: something has gone wrong with formatting here

103: “is used since then” -> “has been widely adopted since then” (recognising there have been a few people who have held out for retaining the term SLI!).

106: “did no longer preclude” -> “no longer precluded”

110-111: the assertion that DLD is now used more widely than SLI is undoubtedly true, but might benefit from adding some numbers? It should be fairly simple to do a count of the number of articles with each term for, say, 2007-2014 vs 2015-2022.

115: realized -> conducted

241: please add a reference for the R programming language

271: “consented results” and “consented terminology” : this is a bit unclear to me. Presumably this refers to the final set of results representing best consensus?

394: ”comprehensive” is rather unclear here – is there an alternative term that could convey what is meant?

489: “rated against” -> “were opposed to”

745: “dissensus” -> “disagreement”

Reviewer #3: This is a very interesting article with a consistent methodology, an ample sample, and a clear and detailed description of the procedure carried out. The results are very well organised, and the conclusions are based on them. My only suggestion refers to the content of Figure 1 (which could be renamed Table 1): The number of experts corresponding to the three categories should be included in each round of the Delphi procedure.

6. PLOS authors have the option to publish the peer review history of their article (what does this mean?). If published, this will include your full peer review and any attached files.

Reviewer #1: No

Reviewer #2: **Yes: **Dorothy V. M. Bishop

Reviewer #3: No

---

## [Author Response · Author response to Decision Letter 0]

11 Oct 2023

Dear Stefano Federici, 

dear Dorothy Bishop, 

We sincerely thank you and the two further, anonymous reviewers for your thorough review of our manuscript and the numerous positive comments you provided. We have meticulously examined all the editorial remarks and the reviewers' suggestions for improving the manuscript, and we have made revisions accordingly. In the following sections, we will address each raised point in detail.

The manuscript meets PLOS ONE’s style requirements.

Since we were funded by a nonprofit organization (GISKID) and not by a typical third party foundation, we are not able to write the correct information in the boxes provided. We would like to give the following information:

This research was funded by the Society for Interdisciplinary Language Acquisition Research and Child Language Disorders in the German-Speaking Countries (GISKID) to the whole D-A-CH Konsortium SES (DACH 2020/1-1; www.giskid.eu). We acknowledge support from the Open Access Publication Fund of the University of Muenster and the University Hospital Münster.

CL is the first chair of GISKID and was at the same time intensively involved in the planning and execution of the study and together with CK mainly responsible for the composition of the publication. 

3. We noted in your submission details that a portion of your manuscript may have been presented or published elsewhere. [A German article on the Delphi study will be published in Logos for the use of German-speaking practitioners, while the current manuscript is aimed at the international scientific community.] Please clarify whether this publication was peer-reviewed and formally published. If this work was previously peer-reviewed and published, in the cover letter please provide the reason that this work does not constitute dual publication and should be included in the current manuscript.

This point had already been clarified intensively in advance with the editorial office. There is a detailed explanation on our part in the online system (including a translation of the German paper and a detailed list of differences between both papers). Subsequently, we received a positive feedback from the editorial office via email, whereupon the review process was opened. Dorothy Bishop (Rev. #2) also explicitly agrees with our argumentation in her review that in this case an international publication, in addition to a German-language publication is very appropriate: “It is an excellent idea to publish an English translation of the outcome, to facilitate comparisons across languages. This is a major study that should have real impact in streamlining research in German-speaking countries and facilitating comparisons with English-speaking groups.”

We checked the reference list and can ensure that it is complete and correct. As indicated by Reviewer 2, we added three references ([4], [24], and [56]). Therefore, all other references changed in their numbers, but no further changes were made.

Reviewers' comments:

Reviewer's Responses to Questions

Comments to the Author

1. Is the manuscript technically sound, and do the data support the conclusions?

Reviewer #1: Yes

Reviewer #2: Yes

Reviewer #3: Yes

2. Has the statistical analysis been performed appropriately and rigorously? 

Reviewer #1: Yes

Reviewer #2: Yes

Reviewer #3: N/A

3. Have the authors made all data underlying the findings in their manuscript fully available?

Reviewer #1: Yes

Reviewer #2: No

Reviewer #3: Yes

The questionnaires and data from all three survey rounds are available on Open Science Framework: https://doi.org/10.17605/OSF.IO/6MB2Pll

We specified this only within the designated section during the submission process. We have now included this information at the end of the methods section as well.

4. Is the manuscript presented in an intelligible fashion and written in standard English?

Reviewer #1: Yes

Reviewer #2: Yes

Reviewer #3: Yes

5. Review Comments to the Author

Reviewer #1: The authors report on the results of a delphi study on the definition and terminology of language disorders in childhood.

This is an important topic and the authors report convincingly the history and background of this endeavor. The experts involved represent all relevant professional fields. Results are clearly presented and discussed. It is also clearly stated when no consensus could be reached. Strengths and limitations are well discussed.

Thank you very much for your positive feedback!

Two thoughts:

- Statement 9. Multilingual children can only be classified as having a developmental language disorder if they do not demonstrate age-appropriate abilities in any of their spoken languages. This sentence is difficult to understand, even with the paragraph explaining it. Could the authors add a sentence such as ‘this means that multilingual children cannot be classified as having a developmental language disorder if they are age-appropriate in one of their spoken languages’ –following ‘Participants agreed on this statement with a mean response of 2.4 (SD = 2.0).’

Thank you for this suggestion. We added the sentence as proposed.

- The aspect of difficulties in nonverbal intelligence together with a developmental language disorder is of high importance. It is interesting to read that no consensus regarding the role of nonverbal cognitive abilities in developmental language disorders (whether a potentially co-causing factor) was reached when considering IQ-score ranges and, thus, no recommendation for the consideration of defined IQ-score ranges of nonverbal cognitive abilities in developmental language disorders was given. But could the authors add a sentence that an evaluation of cognitive abilities is of high importance in a child with language delay – as is also discussed later (lines 665-666).

We added a sentence as recommended.

Reviewer #2: It’s perhaps not surprising that I’ve got very few comments, given that this is a report of a Delphi project that was carefully planned and co-ordinated with many co-authors, and the paper is a straightforward account of the final statements.

The aim of the study was to use the Delphi method to create an agreed definition and terminology for childhood language disorders in German-speaking languages. It is an excellent idea to publish an English translation of the outcome, to facilitate comparisons across languages. This is a major study that should have real impact in streamlining research in German-speaking countries and facilitating comparisons with English-speaking groups.

Thank you very much for your supportive feedback!

As with the CATALISE English-language study, a wide range of professional groups was included, ranging from those with medical, educational and linguistics expertise.

The one limitation, compared to CATALISE, was the lack of any representation from charities working with language-impaired groups – I suspect these may not exist in the German-speaking countries. In general, it can be good to include the ‘user perspective’, but it is admittedly difficult when the condition under consideration is language disorder, and the discussions are all about which words to use. But this would be worth a mention.

Yes, this is absolutely true. There are no charities working specifically with children or families with children with developmental language disorder in the German-speaking countries. We tried to include families in the study, but based on this situation we were not successful. As a result the “Gesellschaft für interdisziplinäre Spracherwerbsforschung und kindliche Sprachstörungen im deutschsprachigen Raum” (GISKID) [Society for Interdisciplinary Language Acquisition Research and Child Language Disorders in the German-speaking Countries] has been making intensive efforts since the end of 2022 to support the establishment of a self-advocacy group for parents of children with developmental language disorders in the German-speaking countries. 

We added a mention, that it was not able to include the perspective of families. 

I could see no mention of open data. It would be useful to make the anonymised reports from the panel openly available; I suspect these won’t attract much interest currently, but when the history of children’s language disorders comes to be written, such information will be invaluable.

The questionnaires and data from all three survey rounds are available on Open Science Framework: https://doi.org/10.17605/OSF.IO/6MB2Pll

We specified this only within the designated section during the submission process. We have now included this information at the end of the methods section as well.

It was good to see broad agreement between this German-language Delphi and the English-language CATALISE. It might be worth saying a little more about the recommendation re using standardized tests to define disorder. This, of course, has the limitation that it requires adequately standardized tests, which may not be available in all countries. I wondered if it might be worth adding an Appendix with a few guidelines regarding suitable standardized tests?

We have incorporated a short paragraph into the discussion on this (l. 706-711), where we also refer to a publication by Spreer (2018). This publication offers a systematic overview of the available German tests. We have refrained from listing them here as an appendix, since a list of German-language procedures would only appeal to a comparatively small number of readers, and we do not feel able to make a recommendation that would apply across several countries and languages.

I think it would be helpful to have a little table at the end that lists the points where the D-A-CH SES consortium and CATALISE consortium diverge. This could help set an agenda for future research that might bring different language definitions into closer alignment.

We added Table 1 with the divergent results of CATALISE and D-A-CH Konsortium SES

Finally, it might be worth inventing an acronym to make it easy to refer to this study!

“D-A-CH Konsortium SES” was our variant of an acronym to refer to the study. It stands for D: Germany, A: Austria, CH: Switzerland, SES: Sprachentwicklungsstörung [developmental language disorder]. It is not quite as catchy as CATALISE, but since we already have it well established in the German-speaking countries, we would stick with it here in the English-language publication.

**Minor wording suggestions**

87 dispute -> debate

Corrected.

88-89: the journal is the International Journal of Language and Communication Disorders, and the summary article would be worth citing here:

Reilly, S., Bishop, D. V. M., & Tomblin, B. (2014). Terminological debate over language impairment in children: Forward movement and sticking points. International Journal of Language & Communication Disorders, 49(4), 452–462. https://doi.org/10.1111/1460-6984.12111

We are sorry about this mistake, corrected it an added the reference.

101-102: something has gone wrong with formatting here

Corrected.

103: “is used since then” -> “has been widely adopted since then” (recognising there have been a few people who have held out for retaining the term SLI!).

Corrected.

106: “did no longer preclude” -> “no longer precluded”

Corrected.

110-111: the assertion that DLD is now used more widely than SLI is undoubtedly true, but might benefit from adding some numbers? It should be fairly simple to do a count of the number of articles with each term for, say, 2007-2014 vs 2015-2022.

Yes, that what we did actually for 2012-2023 (six years before the publication of CATALISE2, and six years after it), but did not report appropriately. We added the numbers now.

115: realized -> conducted

Corrected.

241: please add a reference for the R programming language

We added the reference.

271: “consented results” and “consented terminology” : this is a bit unclear to me. Presumably this refers to the final set of results representing best consensus?

Yes, this refers to the final set of results representing the statements and terminology, for which the previously defined consensus criteria have been met. We rewrote the sentences accordingly. 

394: ”comprehensive” is rather unclear here – is there an alternative term that could convey what is meant?

We changed it to “more complex”. 

489: “rated against” -> “were opposed to”

Corrected.

745: “dissensus” -> “disagreement”

Corrected.

Reviewer #3: This is a very interesting article with a consistent methodology, an ample sample, and a clear and detailed description of the procedure carried out. The results are very well organised, and the conclusions are based on them. My only suggestion refers to the content of Figure 1 (which could be renamed Table 1): The number of experts corresponding to the three categories should be included in each round of the Delphi procedure.

Thank you for your positive feedback! We checked the guidelines for figures and tables, provided by PLOS ONE. In our understanding our visualization of the Delphi process shall be designated as figure. Maybe the academic editor or the editorial office should give us feedback about this matter. 

We wanted to integrate the number of experts corresponding to the three areas (research, teaching, and clinical practice) in the three rounds of data collection, but since many experts were working in two (or even all three) areas, the total number would be different to the number of participants and therefore very confusing in the figure. Therefore, we did not include this information. There were no changes with regard to the portion of participants working in the three areas over the course of the three delphi-rounds. We hope that the information given in line 199-202 represents the required information to a sufficient extent: In all three rounds, most of the respondents stated that they were currently working in the practical field (around 80%). About 30% of the respondents stated that they were additionally or exclusively active in research and about 40% additionally or exclusively in teaching.).

---

## [Decision Letter · Decision Letter 1]

19 Oct 2023

Definition and terminology of Developmental Language Disorders - Interdisciplinary consensus across German-speaking countries

PONE-D-23-16829R1

Dear Dr. Lüke,

We’re pleased to inform you that your manuscript has been judged scientifically suitable for publication and will be formally accepted for publication once it meets all outstanding technical requirements.

Kind regards,

Stefano Federici, Ph.D.

Academic Editor

PLOS ONE

Additional Editor Comments (optional):

Reviewers' comments:

Reviewer's Responses to Questions

**Comments to the Author**

1. If the authors have adequately addressed your comments raised in a previous round of review and you feel that this manuscript is now acceptable for publication, you may indicate that here to bypass the “Comments to the Author” section, enter your conflict of interest statement in the “Confidential to Editor” section, and submit your "Accept" recommendation.

Reviewer #1: All comments have been addressed

Reviewer #2: All comments have been addressed

Reviewer #3: All comments have been addressed

2. Is the manuscript technically sound, and do the data support the conclusions?

Reviewer #1: Yes

Reviewer #2: (No Response)

Reviewer #3: Yes

3. Has the statistical analysis been performed appropriately and rigorously? 

Reviewer #1: Yes

Reviewer #2: (No Response)

Reviewer #3: N/A

4. Have the authors made all data underlying the findings in their manuscript fully available?

Reviewer #1: Yes

Reviewer #2: (No Response)

Reviewer #3: Yes

5. Is the manuscript presented in an intelligible fashion and written in standard English?

Reviewer #1: Yes

Reviewer #2: (No Response)

Reviewer #3: Yes

6. Review Comments to the Author

Reviewer #1: (No Response)

Reviewer #2: (No Response)

Reviewer #3: The authors have adequately addressed the reviewers' concerns and made the manuscript acceptable for publication.

7. PLOS authors have the option to publish the peer review history of their article (what does this mean?). If published, this will include your full peer review and any attached files.

Reviewer #1: No

Reviewer #2: **Yes: **Dorothy V M Bishop

Reviewer #3: No

---

## [Editor Report · Acceptance letter]

25 Oct 2023

PONE-D-23-16829R1 

Definition and terminology of Developmental Language Disorders - Interdisciplinary consensus across German-speaking countries 

Dear Dr. Lüke:

I'm pleased to inform you that your manuscript has been deemed suitable for publication in PLOS ONE. Congratulations! Your manuscript is now with our production department. 

Kind regards, 

on behalf of

Prof. Stefano Federici 

Academic Editor

PLOS ONE